# Classification of WHO Essential Oral Medicines for Children Applying a Provisional Pediatric Biopharmaceutics Classification System

**DOI:** 10.3390/pharmaceutics11110567

**Published:** 2019-10-31

**Authors:** Jose-Manuel delMoral-Sanchez, Isabel Gonzalez-Alvarez, Marta Gonzalez-Alvarez, Andres Navarro, Marival Bermejo

**Affiliations:** 1Department of Pharmacokinetics and Pharmaceutical Technology, Miguel Hernandez University, 03550 San Juan de Alicante, Spain; jmoral@goumh.umh.es (J.-M.d.-S.); marta.gonzalez@goumh.umh.es (M.G.-A.); mbermejo@goumh.umh.es (M.B.); 2Institute of Molecular and Cellular Biology of Miguel Hernandez University, 03202 Elche, Spain; 3Pharmacy Service, General University Hospital of Elche, 03203 Elche, Spain; navarro_and@gva.es

**Keywords:** pediatrics, biopharmaceutics, BCS, solubility, permeability, security

## Abstract

The objective was using the Essential Medicines List for children by the World Health Organization (WHO) to create a pediatric biopharmaceutics classification system (pBCS) of the oral drugs included in the Essential Medicines List by the World Health Organization and to compare our results with the BCS for adults (aBCS). Several methods to estimate the oral drug dose in different pediatric groups were used to calculate dose number (Do) and solubility (high/low). The estimation of the gastrointestinal water volume was adapted to each pediatric group. Provisional permeability classification was done by comparison of each drug lipophilicity versus metoprolol as the model drug of high permeability. As a result, 24.5% of the included drugs moved from the favorable to unfavorable class (i.e., from high to low solubility). Observed changes point out potential differences in product performance in pediatrics compared to adults, due to changes in the limiting factors for absorption. BCS Class Changes 1 to 2 or 3 to 4 are indicative of drugs that could be more sensitive to the choice of appropriate excipient in the development process. Validating a pBCS for each age group would provide a valuable tool to apply in specific pediatric formulation design by reducing time and costs and avoiding unnecessary pediatric experiments restricted due to ethical reasons. Additionally, pBCS could minimize the associated risks to the use of adult medicines or pharmaceutical compound formulations.

## 1. Introduction

The objective of this paper was using the Essential Medicines List for children by the World Health Organization (WHO) to create a provisional pediatric biopharmaceutics classification system (pBCS) for oral drugs and to compare our results with the BCS for adults (aBCS).

Since 2007, the WHO has provided the WHO Model List of Essential Medicines for Children [1]; a core drug list for a basic health-care system. This resource list considers the safest, efficacious and cost-effective drugs for priority pathologies in pediatrics up to 12 years old [1]. All immediate release oral drugs from the sixth edition of this list were selected for this work.

The need for and importance of developing appropriate, safe and effective medicines for pediatrics have now been recognized. Pediatrics have long been considered as a therapeutically disregarded group [2]. This fact lies in the therapeutic gap that originates from formulation development (mainly focused on adults), low availability of commercial pediatric drugs for pediatric use and physiological and anatomical differences relative to the adult population.

In younger pediatrics, neonates and infants, the therapeutic-risk benefits associated with drug treatment may be different from those in adults. Developmental physiological differences (as the composition of gastrointestinal fluids and the presence or activity of intestinal drug metabolizing enzymes and efflux transporters) can markedly alter the bioavailability of drugs [3]. Age-related changes in the characteristics of luminal fluids such as pH may lead to changes in drug solubility and, therefore, in absorption since only the dissolved drug would be available for absorption [4,5]. Since fluctuations in pH may influence ionizable drugs and the relative amount of unionized available drug, there could be changes in absorption during maturation [3]. The gastric pH of the newborn is approximately neutral, and within the first 48 h of life, the gastrointestinal pH decreases to adult values and then for the first month increases back to values close to neutrality. After the first month, gastric pH decreases progressively to reach similar values as in adults by two years of age [4,6,7].

On the other hand, another determinant of bioavailability is ontogenic changes in the expression of drug metabolizing enzymes and transporters [6,8], which would mark the first pass effect and consequently would change the oral absorbed fraction. For example, the CYP3A4 isoenzyme increases expression and activity with age [9]. CYP3A4 is practically undetectable in neonates, and its expression increases with age, reaching a maximum level at two years of age [6,9]. Likewise, in these age groups, there is immature intestinal activity of alkaline phosphatase, glucuronidation enzymes and carboxylesterase-2. Conversely, there are no differences in p-glycoprotein expression, the most studied intestinal efflux transporter [6] from the earliest stages of life to adults [6].

Many drugs are formulated as pharmaceutical forms that are often not appropriate for pediatrics, such as tablets or capsules [10]. Children need different oral dosage formulations from adults mainly due to their developmental capabilities in swallowing ability, palatability and dosage requirements. Especially in pediatrics, palatability is closely related to acceptance and adherence. Consequently, bitterness, trigeminal irritation and perceptible malodors must be minimized [11,12].

The increasing need for age-related medicines in pediatrics could be met by extemporaneous formulations. The compounding of an extemporaneous formula is not allowed if an authorized commercial pharmaceutical product (adequate for a particular age group) exists [13], but the reality in the clinical setting is that for some age groups, that is not the case, while extemporaneous formulation is still used as the only alternative to the therapeutic need [7,14]. Although solid products have higher stability than liquid formulations [15], this practice can solve the problem of swallowing capability and dose adjustments, but these manufactured preparations can be a risk to safety, efficacy and quality, because there is little or no information for compatibility between the original medicine and additional excipients or their effect in children [16]. The excipients used in formulations designed for children should be appropriate for each age group [15] because, as is known, these are not inert substances [16] and cannot be supposed to have the same effects in pediatrics as in adults. The most common example is early stage children who are not able to metabolize or eliminate them as adults [17,18]. The ideal situation in the development of pediatric formulations is the minimal use of added excipients, both in number and quantity [8].

The Biopharmaceutics Classification System (BCS) has been established as the predictor of drug absorption in pharmaceutical development, simplifying approval and drug development processes [19]. BCS considers the fundamental parameters controlling the absorption of any drug administered orally [20], drug solubility and drug permeability, and it classifies the drugs into four groups (Class 1: high permeability, high solubility; Class 2: high permeability, low solubility; Class 3 low permeability, high solubility; and Class 4: low permeability, low solubility). The BCS class in combination with the drug product dissolution rate determines the feasibility of granting biowaivers, i.e., the possibility of demonstrating bioequivalence with in vitro dissolution studies [21,22]. A drug is considered highly permeable when the amount of drug absorbed is greater than 85–90% of the total administered dose [21,22]. High solubility drugs are those in which the highest strength (FDA) or highest single dose (EMA) is soluble in 250 mL of aqueous media (pH range 1–6.8) derived from typical bioequivalence study protocols [23].

Children’s idiosyncrasies and physiological differences from adults justify the need for a specific pediatric Biopharmaceutics Classification System (pBCS), preferably by age group. The main reasons for the need for a pediatric BCS are to promote pediatric-fitted formulations’ development and to try to assess the risk of the inadequacy of an adult formulation to pediatrics [24]. Furthermore, a pediatric pBCS could help to identify the risks associated with extemporaneous formulation as low solubility drugs would be in particular sensitive to formulation differences, highlighting the relevance of standardized formulations across health institutions to ensure consistent performance.

For all the above mentioned reasons, the development of a pediatric BCS could improve pediatric drug product development. Moreover, this classification system endeavor will be more complicated than an adult BCS due to the complexity of pediatric groups. In the present work, we attempted a provisional pBCS classification to explore how many drugs could potentially present challenges in some pediatric groups and thus being candidates for harmonized extemporaneous formulations in the hospital setting and to further develop age appropriate oral formulations.

## 2. Materials and Methods

### 2.1. Pediatric Solubility Classification

Pediatric drug solubility classification was determined via dimensionless pediatric dose number calculation with the following equation [25]:*D*_0p_ = (*M*_0p_/*V*_0p_)/*C*_s_
where *M*_0p_ is the pediatric highest dose (in milligrams), *V*_0p_ is the pediatric reference volume (predicted water volume in milliliters taken with a dose) and *C*_s_ is the solubility of the drug in water in milligrams per milliliter.

Highly soluble drugs were considered those with calculated dose number ≤ 1.

Unlike the adult classification, there would not be a single pBCS classification. Each subgroup used would provide a different dose number because dose and pediatric reference volume are age-related factors.

Three different pediatric subgroups were considered [24,26,27]: neonate (0–1 month), infant (1–24 months) and child (2–12 years). For each group, the volume was estimated based on the volume of gastric fluids in fasted children, approximately 0.56 mL/kg [28,29,30], relative to the gastric volume of a 70 kg fasted adult (37.1 mL) [29,30] and BCS adult volume of 250 mL. A 0.56 mL/kg value was applied to all the studied groups as a conservative estimation. This estimation could be improved in the future when better estimations of fluid intestinal content become available. Subgroup *V*_0_ was estimated from:*V*_0_ (mL) = [[0.56 mL/kg × weight (kg)]/37.1 mL]/250 mL
where weight was taken to be the 50^th^ percentile of boy values in the WHO growth charts [31] for neonates (mean age: 0.5 months) and infants (mean age: 12.5 months) and the Center of Disease Control and Prevention Charts [32] for children (mean age: 7 years).

Referenced *M*_0p_ for each age group was obtained from the British National Formulary of Children [33], WHO Model Formulary for Children [34], WHO Model Prescribing Information: Drugs Used in Parasitic Diseases [35] and, conversely, via common clinical equations [36]: Fried’s Rule (FR) (in neonates), Clark’s Rule (CR) (in infants and children) and Body Surface Area method (BSA) (in all groups).

Fried’s Rule: [age (months)/150] × adult dose

Clark’s Rule: [weight (pounds)/150] × adult dose

BSA method: [BSA patient (m^2^)/1.73 m^2^] × adult dose

For dose calculation with the BSA method and for referenced doses expressed in mg/kg, the same resources were used as for calculating the reference volume [31,32]. The referenced dose was considered as the gold standard because it is based on clinical use. Alternative dosages were included since some drugs do not have an associated dose in the selected literature.

Drug solubility values (milligrams per milliliter) were obtained from The International Pharmacopoeia [37] and Merck Index [38]. The most conservative values were applied. The estimated lower solubility limit of the values range defined in USP [39] according to Table 1 were necessary in drugs with an unavailable solubility specific value.

### 2.2. Permeability Classification

For permeability value assignment, passive transcellular diffusion was considered as the single transport mechanism across biological membranes, dismissing the possible carrier-mediated mechanisms [41]. Therefore, permeability classification was based on partition coefficient, log*P* (n-octanol/water partition coefficient) and its relationship with human intestinal permeability. Log*P* values were obtained from the Chemicalize platform [42].

Metoprolol was selected as the benchmark between low or high permeable drug (log*P* 1.76) [40,43,44]. Log*P* values lower than 1.76 were associated with low permeability, and log*P* values larger than or equal to 1.76 were associated with high permeability.

## 3. Results

### 3.1. Classification of Drug Solubility

The common estimated parameters for all included drugs to calculate the solubility are displayed in Table 2.

Figure 1 shows the percentage of drugs with respect to the total that resulted in being highly soluble.

### 3.2. Classification of Drug Permeability

Similar to other studies [40,43,44], the classification was based on the relationship between log*P* values and human intestinal permeability. Since drugs with estimated log*P* values greater than or equal to 1.76 were rated as high permeability drugs, 59 of the drugs selected (41.3%) were assigned to this group. The remaining 84 drugs (58.7%) were classified as low permeability class.

### 3.3. Provisional Pediatric Biopharmaceutics Classification System (pBCS) Classification of the WHO Essential Oral Medicines for Children

The sixth version of the WHO Essential Oral Medicines for Children provides 143 drugs for oral administration. All these drugs were classified according to a provisional pBCS. Figure 2, Figure 3 and Figure 4 show the provisional pBCS distribution for each pediatric subgroup, according to each dosage method.

Table A1 found in Appendix A summarizes our provisional pBCS classification of the orally administered Essential Medicines for Children according to their log*P* and the calculated *D*_0p_ using the pediatric reference volume for each subgroup and different dosages.

### 3.4. Comparing Provisional Pediatric Biopharmaceutics Classification System (pBCS) with BCS in Adults

Table 3 shows drugs with unfavorable changes to BCS adults, and Table 4 shows favorable changes. Unfavorable changes were defined as those changes that involve a switch to a low solubility class (1 to 2 or 3 to 4) from adult BCS (aBCS) to pBCS. Inversely, class switches to a high solubility class (2 to 1 or 4 to 3) from aBCS to pBCS were defined as favorable changes.

## 4. Discussion

### 4.1. Drug Solubility Classification

Drug solubility, highest dose and volume are parameters that vary among all pediatric groups and as a consequence may change the solubility classification [45].

It is important to note that the BCS is based on adult fasted states, and these conditions are difficult to achieve in neonates and the youngest infants due to breastfeeding frequency or formula meals [45]. Instead, Hens et al. [46] found in their study that the youngest children preferred liquid formulations as solutions or suspensions, which do not require extra liquid for swallowing. In this case, the available liquid for drug dissolution would be the volume provided by the liquid formulation plus the one available in fasted or fed stomach [45,46].

Solubility is a critical factor since the dissolved drug is the one that is available for absorption and would determine bioavailability. It is important to note that solubility widely increases with temperature. Solubility data used in this study correspond to calculations performed at 25 °C; body temperature may increase drug solubility, and for that reason, this classification could be considered on the conservative side.

In addition, gastrointestinal fluid composition may lead to changes in drug properties like solubility [47], affecting directly poorly soluble drugs (BCS Class 2 and 4 drugs). Compared to adults, neonates are the pediatric subgroup with more important composition differences such as pH, presence of milk/formula or bile salt concentration [47].

In this study, as well as for adults, dose number was used to estimate solubility class, but varying the volume according to age ranges. Since the permeability classification in the adult and pediatric subjects has been compared in relation to the partition coefficient log*P*, the determinant factor of a change in the pBCS in the present study is the estimated dose number. It has been suggested that using reduced volumes and reduced doses in pediatrics compared to adult values, the effect on the BCS classification is minimal [30,48]; therefore, it should not be affected in most drugs. In the present study, 24.5% of drugs showed an unfavorable change in BCS class, which is not a negligible percentage.

In regard to solubility between age groups (Figure 1), in infants (12.5 months) and children (seven years), there was more homogeneity among dosages than in neonates (0.5 months). In neonates, dosage via Fried’s rule provided lower doses compared to BSA dosage or the reference dose, identifying drugs as highly soluble and, perhaps, providing sub-therapeutic treatment. Additionally, this dose underestimation caused the majority of favorable changes in pBCS relative to aBCS. Low calculated doses resulted in a low *D*_0p_, and therefore, a pBCS group changes to another with a favorable solubility classification.

If dosage by Fried’s rule in neonates is not taken into account, all percentages are around 40–50% of highly soluble drugs (Figure 1). This percentage could be the potential drug products that may profit from the biowaiver approach, i.e., demonstrating bioequivalence with in vitro dissolution bioequivalence tests. If those drugs are candidates for a waiver of in vivo bioequivalence testing, the development process is facilitated much more in pediatrics in which the ethical implications of any clinical study are a great concern. Unfortunately, this percentage was lower than that the estimated for adults (40–50% vs. 67% [40]) by Kasim et al. [40].

A relevant limitation of this study related to BCS classification in children is the lack of consensus about the reference volumes of each pediatric subgroup and the volume of liquid taken with medicines [29]. Furthermore, the water amount taken is a conservative factor because: (1) it is an estimate of the fluid volume available in the gastrointestinal tract under the fasted state condition [27]; and (2) neonates and the youngest children are in a fed state most of the time due to the higher frequency of food intake according to breastfeeding [45,47].

There are currently three methods to calculate the reference volume in the bibliography: (1) using a fixed value of 25 mL for all age groups [8], (2) using a related volume to BSA [24,36] and (3) using fasted gastric volumes linked to body weight [4,30], this latter method having been used in the present study due to it being the most conservative. This fact emphasizes the need for additional research on gastrointestinal pediatric fluid volumes.

Drugs with unfavorable changes in solubility classification are those that originated the changes in the pBCS classification (Table 3), as the permeability classification was considered equal in adult BCS. This aspect is also a limitation of this study as eventually, intestinal permeability may also change with maturation. For example, molecules that undergo a change from 3 to 4 in all stages and with dosing by the referenced dose method are acetylsalicylic acid, amoxicillin or nystatin. Mefloquine and omeprazole experienced a change from 1 to 2.

Furthermore, additional consideration for pBCS should be given to the fact that there are pediatric subgroups with a wide range of age. The estimation of doses and volumes with the means of the age of each subgroup was considered as a limitation of the present study because these calculations might not represent the entire group.

### 4.2. Drug Permeability Classification

There is limited biopharmaceutical and pharmacokinetic information on drug permeability in pediatrics [30,45], especially in early ages, which is an under researched area [49]. Due to this deficiency, alternative methods are needed, as for instance to correlate permeability with physicochemical parameters such as log*P* to allow for a provisional estimation and physiologically based pharmacokinetics (PBPK) modelling [45].

It is generally accepted that the greatest differences between adults and children are found in children less than two years of age [30,50]. During development, permeability changes progressively from birth to two years of age, when it is considered equivalent to the permeability data in adults [4]. Younger children have a great component of paracellular absorption [30], and this mechanism has been ignored in the present study as stated above, so we consider this fact as a limitation. Most of the selected drugs (58.7%) were classified as low permeability drugs with the used method based on log*P*.

Currently, pharmaceutical companies use biopharmaceutics to predict in vivo performance in medicines’ development. This design method by biopharmaceutical tools is extensively explored for adults, but not for pediatrics [51]. As explained above, pediatric subjects are not small adults. Age related physiological and anatomical changes affect drug absorption distribution, metabolism and excretion [52], so it is essential to characterize those changes to get accurate pharmacokinetic predictions in all age groups.

Three main biopharmaceutical tools are presently recognized as essential elements for drug development and part of regulatory submissions: PBPK models (in silico), dissolution tests of the formulation (in vitro) and BCS classification. Extrapolating of aBCS is not a straightforward task, and the predictive capability of PBPK for pediatrics is still in its initial stages of development, although it is fortunately rapidly evolving [50,52] since realistic levels of variability within each age group are incorporated into the tool [53]. The main reason for the low predictive capability is mainly due to a lack of validated pediatric PBPK models for oral drug absorption and disposition [50,54]. Despite this, developing drug formulations by these models will undoubtedly play a critical role.

For example, Khalil et al. [54] studied the use of PBPK models with sotalol in all pediatric stage groups by applying two different models: Simcyp^®^ (Simcyp Ltd., Sheffield, U.K.) and PK-SIM^®^ (Bayer Technology Services GmbH, Leverkusen, Germany). They concluded that the lower predictive performance was seen in neonates; in contrast with the other pediatric groups with good model predictability. In this respect, the youngest groups are in general the most disadvantaged due to their bigger and still not well characterized physiological and anatomical differences. Developing this line of biopharmaceutical research would help the development process of specific pediatric formulations.

PBPK would be the optimal tool for permeability prediction in pediatric patients. In addition, a validated pBCS system would facilitate formulation development, as the rate limiting factors would be identified thus the adequate pharmaceutical technology that could be used to overcome them. In parallel formulation comparison would be made with the adequate tool either in in vivo bioequivalence trials or in in vitro dissolution studies.

### 4.3. Provisional pBCS Classification

Considering drug solubility, permeability based on the log*P* method and highest doses administered, all 143 WHO Essential Oral Drugs for pediatrics were classified into a provisional pBCS.

The distribution of BCS classes in the 143 studied drugs according to the reference dosage method (Figure 2, Figure 3 and Figure 4) was 12.9%/14.3%/16.1% (neonates/infants/children) for Class 1, 20.0%/26.8%/32.1% for Class 2, 30.0%/26.8%/31.3% for Class 3 and 37.1%/32.1%/33.0% for Class 4. Homogeneity in BCS class according to all considered dosages was found (Table 4), except for Fried’s rule dosage in neonates. In this case, higher percentages of high solubility classes (Classes 1 and 3) were found due to possible underdosing mentioned above.

Considering unfavorable changes (Table 3), 24.5% of classified drugs modified their class between aBCS and pBCS to an unfavorable class. Furthermore, 77.1% of these drugs with unfavorable changes showed a switch in reference dose in all stage groups.

When an extemporaneous formulation for pediatrics in the hospital setting is unavoidable due to the lack of an adequate commercial authorized one, excipients could affect formulation performance for Class 2 and 4 drugs for which solubility and dissolution are the limiting factors affecting fraction absorbed. In those cases, different formulations in different hospitals may have different rates and extents of absorption. The clinical relevance of such differences will depend on the particular drug, but this potential risk could be avoided with the harmonization of the compound formulas across institutions.

Observed changes point out also potential differences in product performance in pediatrics compared to adults, due to the change in the limiting factors for absorption. As mentioned, the BCS class changes from 1 to 2 or 3 to 4 are indicative of drugs that could be more sensitive to the choice of appropriate excipient in the development process. Validating a pBCS for each age group would provide a valuable tool to apply in specific pediatric formulation design by reducing time and costs and avoiding unnecessary pediatric experiments restricted by ethical reasons. With a validated pBCS, the biowaiver approach, i.e., the demonstration of formulations’ bioequivalence by in vitro dissolution studies, would be of application, as it is currently used in medicines for adults. Additionally, pBCS could minimize the associated risks to the use of adult medicines on pharmaceutical compound formulations for children.

## 5. Conclusions

On average, 24.5% of the 143 drugs evaluated in the present study modified their class between aBCS and the proposed pBCS to an unfavorable class (i.e., from high to low solubility). Even if the proposed pBCS would need further refinement, this percentage is not negligible.

Research in neonates and younger infants needs to be prioritized because there is less certainty of our knowledge about their gastrointestinal physiological factors affecting oral absorption. The great physiological differences between adults and youngest children place this group in a weaker position where predictive modelling from adults is still inadequate and the access to clinical research is limited due to ethical barriers. Pediatric biopharmaceutics and a validated pBCS can be risk-assessment tools in the necessary optimization tasks for developing age appropriate oral medicines. Due to the potential changes found in the present study regarding adults, developing a validated pBCS would help to improve the safety of pediatric therapeutics.

## Figures and Tables

**Figure 1 pharmaceutics-11-00567-f001:**
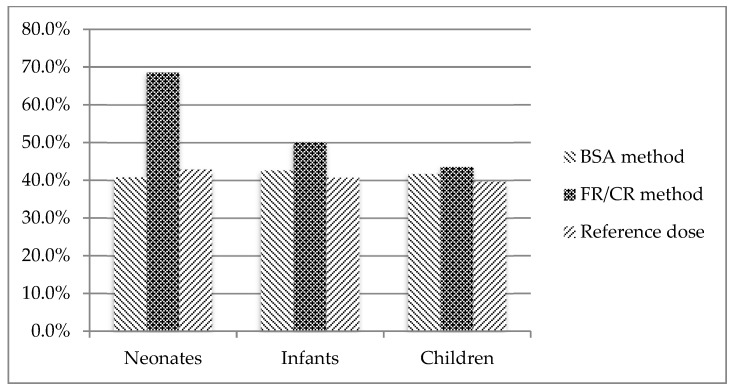
Distribution of high solubility drugs for each age group (FR: Fried’s Rule, CR: Clark’s Rule).

**Figure 2 pharmaceutics-11-00567-f002:**
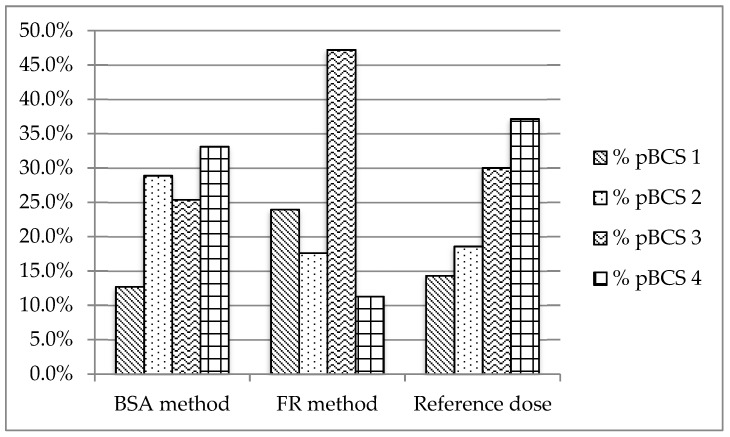
Distribution of provisional pBCS classification in neonates (0–1 month).

**Figure 3 pharmaceutics-11-00567-f003:**
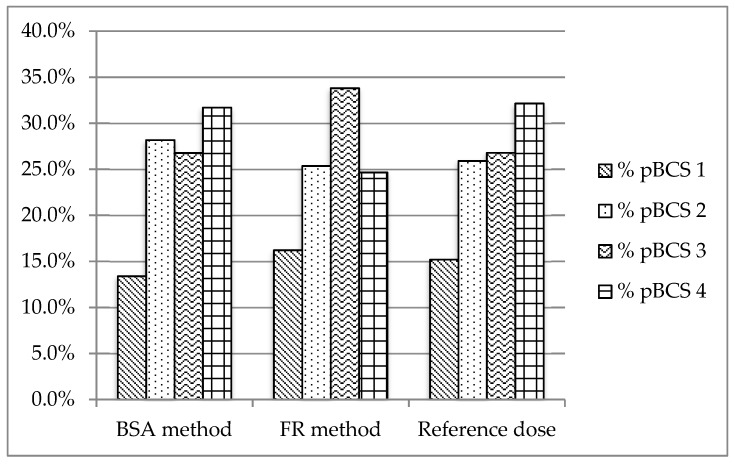
Distribution of provisional pBCS classification in infants (1–24 months)**.**

**Figure 4 pharmaceutics-11-00567-f004:**
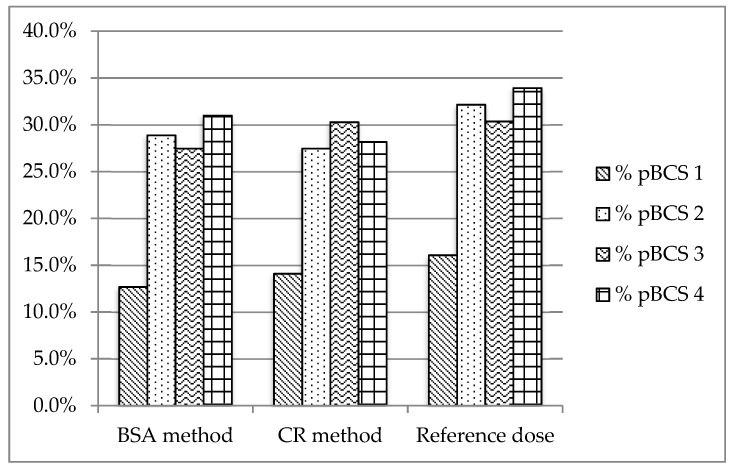
Distribution of provisional pBCS classification in children (2–12 years).

**Table 1 pharmaceutics-11-00567-t001:** Solubility definitions [36,39,40].

Descriptive Term (Solubility Definition)	Parts of Solvent Required for One Part of Solute	Solubility Range (mg/mL)	Solubility Assigned (mg/mL)
Very soluble (vs)	<1	>1000	1000
Freely soluble (fs)	from 1 to 10	100–1000	100
Soluble (s)	from 10 to 30	33–100	33
Sparingly soluble (sps)	from 30 to 100	10–33	10
Slightly soluble (ss)	from 100 to 1000	1–10	1
Very slightly soluble (vss)	from 1000 to 10000	0.1–1	0.1
Practically insoluble (pi)	>10000	<0.1	0.01

**Table 2 pharmaceutics-11-00567-t002:** Shared parameters to obtain the pediatric biopharmaceutics classification system (pBCS) classification. BSA, body surface area.

Parameters	Neonate (0.5 months)	Infant (12.5 months)	Child (7 years)
P50^th^ Weight (kg) [31,32]	3.7	9.8	23
P50^th^ Length (cm) [31,32]	52.5	77	122
BSA (m^2^)	0.23	0.46	0.88
Normalized *V*_0p_ (mL)	13.96	36.98	86.79

**Table 3 pharmaceutics-11-00567-t003:** Provisional pBCS classification of drugs with unfavorable changes with respect to adult BCS (aBCS). Drugs with unfavorable changes with the referenced dose dosage in all age stages are given in bold. Drugs in italics indicate changes in pBCS classification with the referenced dose not in all of the age stages. Abbreviations: BSA-R, body surface dosage; Fr-R, Fried’s Rule dosage; Cr-R, Clark’s Rule dosage; Ref, referenced dosage; aBCS, adult BCS.

Drugs	Provisional pBCS									Provisional aBCS
	Neonate (0.5 months)			Infant (12.5 months)			Child (7 years)			
	BSA-R	Fr-R	Ref	BSA-R	Fr-R	Ref	BSA-R	Cr-R	Ref	
Acetylsalicylic acid	4	4	4	4	4	4	4	4	4	3
Acyclovir	4	3	4	4	4	4	4	4	4	3
Amodiaquine	2	2	-	2	2	2	2	2	2	1
Amoxicillin	4	3	4	4	4	4	4	4	4	3
Benznidazole	4	4	4	4	4	4	4	4	4	3
Calcium gluconate	4	3	4	4	3	4	4	4	4	3
Cephalexin	4	3	4	4	4	4	4	4	4	1
Chloramphenicol	4	3	4	4	3	4	4	4	4	3
Ciprofloxacin	4	4	4	4	4	4	4	4	4	3
Clindamycin	3	3	3	3	3	3	3	3	3	1
Dexamethasone	4	3	-	4	4	4	4	4	4	3
*Digoxin*	1	1	2	1	1	2	1	1	1	1
*Enalapril*	3	3	3	3	3	3	3	3	3	1
Ethambutol	4	3	3	3	3	3	3	3	3	3
Fluconazole	4	3	4	4	3	4	4	4	4	3
Flucytosine	4	3	4	4	3	4	4	3	4	3
Fludrocortisone	3	3	4	3	3	4	3	3	4	3
*Folic acid*	3	3	4	3	3	4	3	3	3	3
Haloperidol	2	1	-	2	1	-	2	1	2	2
Hydrochlorothiazide	4	3	4	4	4	4	4	4	4	3
Hydrocortisone	4	3	4	4	3	3	4	3	3	1
Mefloquine	2	2	-	2	2	2	2	2	2	1
Mercaptopurine	4	4	-	4	4	-	4	4	-	2
Methotrexate	4	4	-	4	4	-	4	4	-	3
Neostigmine	3	3	3	3	3	3	3	3	3	1
Nifurtimox	4	4	4	4	4	4	4	4	4	3
Nystatin	4	3	4	4	4	4	4	4	4	3
Omeprazole	2	1	2	1	1	2	1	1	2	1
Phenobarbital	4	3	4	3	3	4	3	3	4	3
Prednisolone	4	3	-	4	4	4	4	4	4	1
Proguanil	2	1	2	2	1	2	2	2	2	1
Propylthiouracil	4	3	4	4	4	4	4	4	4	3
Pyrazinamide	4	3	-	4	4	4	4	4	4	3
*Quinine sulfate*	2	1	-	2	2	-	2	2	2	1
Riboflavin	4	3	4	4	3	4	3	3	4	3

**Table 4 pharmaceutics-11-00567-t004:** Provisional pBCS classification of drugs with favorable changes with respect to aBCS. Drugs with favorable changes with the referenced dose dosage in all age stages are given in bold. Drugs in italics indicate changes in pBCS classification with the referenced dosage not in all of the age stages. Abbreviations: BSA-R, body surface dosage; Fr-R, Fried’s Rule dosage; Cr-R, Clark’s Rule dosage; Ref, referenced dosage; aBCS, adult BCS.

Drugs	Provisional pBCS									Provisional aBCS
	Neonate (0.5 months)			Infant (12.5 months)			Child (7 years)			
	BSA-R	Fr-R	Ref	BSA-R	Fr-R	Ref	BSA-R	Cr-R	Ref	
Acetylcysteine	4	3	4	4	4	4	4	4	4	4
Allopurinol	4	3	-	4	4	4	4	4	4	4
Artesunate	2	1	-	2	2	2	2	2	2	2
Azithromycin	2	1	2	2	2	2	2	2	2	2
Cefixime	4	3	-	4	4	4	4	4	4	4
Clarithromycin	2	1	2	2	2	2	2	2	2	2
Dapsone	4	3	-	4	4	4	4	4	4	4
Diazepam	2	1	2	2	2	2	2	2	2	2
Diloxanide	2	1	-	2	1	1	2	1	1	2
Doxycycline	4	3	-	4	4	-	4	4	4	4
*Fluoxetine*	1	1	1	1	1	1	2	2	2	2
Furosemide	4	3	4	4	4	4	4	4	4	4
Ivermectin	2	1	-	2	2	-	2	2	2	2
*Levofloxacin*	4	3	-	4	4	-	4	4	3	4
Linezolid	4	3	4	4	3	4	4	4	4	4
Loratadine	2	1	-	2	2	2	2	2	2	2
Morphine	4	3	3	4	4	3	4	4	3	4
Moxifloxacin	4	3	-	4	4	-	4	4	-	4
*Phytomenadione*	2	1	2	2	2	1	2	2	1	2
Pyrimethamine	2	1	2	2	2	2	2	2	2	4
Retinol	2	1	2	2	2	2	2	2	2	2
Tioguanine	4	3	-	4	3	-	4	3	-	4
Trimethoprim	4	3	4	4	3	4	4	4	4	4
Voriconazole	2	1	-	2	2	-	2	2	2	2

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
