# Peer review of "Classification of WHO Essential Oral Medicines for Children Applying a Provisional Pediatric Biopharmaceutics Classification System"

_pharmaceutics, 2019, doi:10.3390/pharmaceutics11110567_

Round 1

Reviewer 1 Report

The manuscript reports on the attempt to provide oral drugs classification in line with the provisional Pediatric Biopharmaceutics Classification System (pBCS) for a selected data set of drugs which are included in the WHO Essential Medicines List. The underlying assumptions and relevant challenges have been reviewed and discussed taking into account three pediatric subpopulations and different approaches for dose calculation.

The topic and data presented are interesting, however, the manuscript needs further editing in order to improve the language and general readability and comprehension.

Introduction is rather lengthy and should be structured around the specific topics of discussion identified. The authors should reconsider if the BCS and BDDCS schemes presented in figures 1 and 2 are necessary or could be omitted.

It is stated as an objective to “explore how many drugs could potentially present challenges in some pediatric groups and thus being candidates to harmonized extemporaneous formulations in the hospital setting.” However, implications related to extemporaneous medicines preparation have not been further discussed.

Materials and Methods

Abbreviations for “Fried’s Rule” and “Clark’s Rule” should be introduced since they firstly appear in the Figure 3 legend.

Table 1 – dashes are missing in the rows 5 and 6 in column 3; also, it is not clear if the stated references (i.e. 36 and 39) are correct for solubility definition

Tables 3 and 4 replicate considerable amount of data which is also presented as Appendix A. The authors should consider to reduce the width of columns in the table in a way that it may be presented in the “portrait” orientation and list the drugs with the “unfavourable”, “favourable” and without change as the subsets in the common table.

Conclusions should be more specific and better aligned with the stated objectives of the study and results obtained.

There is a number of typing and language omissions, which should be corrected.

Author Response

Reviewer 1

The manuscript reports on the attempt to provide oral drugs classification in line with the provisional Pediatric Biopharmaceutics Classification System (pBCS) for a selected data set of drugs which are included in the WHO Essential Medicines List. The underlying assumptions and relevant challenges have been reviewed and discussed taking into account three pediatric subpopulations and different approaches for dose calculation.

The topic and data presented are interesting, however, the manuscript needs further editing in order to improve the language and general readability and comprehension.

Thanks for your comments. Manuscript language has been edited following your suggestion.

Introduction is rather lengthy and should be structured around the specific topics of discussion identified. The authors should reconsider if the BCS and BDDCS schemes presented in figures 1 and 2 are necessary or could be omitted.

The schemes have been omitted and described in the text.

It is stated as an objective to “explore how many drugs could potentially present challenges in some pediatric groups and thus being candidates to harmonized extemporaneous formulations in the hospital setting.” However, implications related to extemporaneous medicines preparation have not been further discussed.

The potential implications have been further clarified I the discussion. The next paragraph has been included.

When an extemporaneous formulation for paediatrics in the hospital setting is unavoidable due to the lack of adequate commercial authorized one, excipients could affect formulation performance for class II and IV drugs for which solubility and dissolution are the limiting factors affecting fraction absorbed. In those cases, different formulations in different hospitals may have different rate and extent of absorption. The clinical relevance of such differences will depend of the particular drug but this potential risk could be avoided with the harmonization of the compounding formulas across institutions.

Materials and Methods

Abbreviations for “Fried’s Rule” and “Clark’s Rule” should be introduced since they firstly appear in the Figure 3 legend.

Abbreviations have been introduced just after the first definition of both rules.

Table 1 – dashes are missing in the rows 5 and 6 in column 3; also, it is not clear if the stated references (i.e. 36 and 39) are correct for solubility definition

Tables 3 and 4 replicate considerable amount of data which is also presented as Appendix A. The authors should consider to reduce the width of columns in the table in a way that it may be presented in the “portrait” orientation and list the drugs with the “unfavourable”, “favourable” and without change as the subsets in the common table.

Suggested changes have been done.

Conclusions should be more specific and better aligned with the stated objectives of the study and results obtained.

As the stated objective was the classification in pBCS and the quantification of the unfavourable changes the next sentence has been added to the conclusions:

On average 24,5% of the 143 evaluated in the present study modify their class between aBCS and the proposed pBCS to an unfavorable class (i.e from high to low solubility). Even if the proposed pBCS would need further refinement, this percentage is not negligible.

There is a number of typing and language omissions, which should be corrected.

Reviewer 2 Report

The authors should justify the use of the term "pediatric" for <12y.  The presented measurements are done in a solid way, but their clinical and scientific conclusions could be more comprehensible The mantra of children as "therapeutic orphans" should not be blindly repeated. The "pediatric population" is more a regulatory construct than a scientifically correct description of young humans. The examples for different isoenzymes in adults and infants and of different gastric pH of neonates and adults apply to very young children, not to an entire "pediatric population". Specifically the EMA dislikes extemporaneous formulations, so the claim about a broad agreement should be relativized The authors' proposed "Pediatric drug solubility classification" may apply to infants, but not to everybody <12-years-old. Conclusions such as "Research in neonates and younger infants needs to be prioritized because they are the most vulnerable pediatric groups." are placative and dramatizing. Why not say that in neonates and young infants there is less certainty of our knowledge?

Author Response

Reviewer 2

The authors should justify the use of the term "pediatric" for <12y. 

We have included the relevant reference from WHO. We are using the classification of age groups by the World Health Organization

The presented measurements are done in a solid way, but their clinical and scientific conclusions could be more comprehensible The mantra of children as "therapeutic orphans" should not be blindly repeated. The "pediatric population" is more a regulatory construct than a scientifically correct description of young humans.

We agree with the reviewer, but the fact is that development of medicines and extemporaneous compounding must comply with the regulations and consequently those age groups are used to define the target populations.

The examples for different isoenzymes in adults and infants and of different gastric pH of neonates and adults apply to very young children, not to an entire "pediatric population".

Those changes are mentioned in the introduction with the adequate reference mentioning the particular age group for which are relevant, we are not affirming they apply to the whole pediatric group

Specifically the EMA dislikes extemporaneous formulations, so the claim about a broad agreement should be relativized

We agree with the reviewer, not only EMA “dislikes” extemporaneous formulations, actually its use is forbidden if a commercial marketed authorized pharmaceutical product adequate for an age group exists but the reality on the clinical setting is that for some age groups that is not the case and extemporaneous formulation is still used as the only alternative to the therapeutic need as it is also recognized by EMA in its reflection paper included as reference.

We have rephrased that on the introduction.

The increasing need of age-related medicines in pediatrics could be met by extemporaneous formulations. The compounding of an extemporaneous formula is not allowed if an authorized commercial pharmaceutical product (adequate for a particular age group) exists, but the reality on the clinical setting is that for some age groups that is not the case and extemporaneous formulation is still used as the only alternative to the therapeutic need.

The authors' proposed "Pediatric drug solubility classification" may apply to infants, but not to everybody <12-years-old.

Exactly, this is why solubility classification has been stratified for the different paediatric age –groups, in particular for neonates, infants and children up to 7 years old

Conclusions such as "Research in neonates and younger infants needs to be prioritized because they are the most vulnerable pediatric groups." are placative and dramatizing. Why not say that in neonates and young infants there is less certainty of our knowledge?

The conclusion has been rephrased following your recommendation

Round 2

Reviewer 1 Report

The authors have made some effort to improve the manuscript, however, there is still need for a thorough language editing. Otherwise, the manuscript is quite interesting and addresses several issues which are an ongoing topic of scientific and regulatory concern.